# Incidence and persistence of asymptomatic *Leishmania* infection among HIV-infected patients in Trang province, Southern Thailand: A cohort study

Lertwut Bualert[1], Toon Ruang-areerate[2], Mathirut Mungthin[2], Saovanee Leelayoova[2], Suradej Siripattanapipong[3], Tawee Naaglor[2], Nattapong Hongsimakul[2], Supicha Sroythong[2], Phakhajee Rattanalertpaiboon[2], Preeyaporn Tulpeng[4], Phunlerd Piyaraj[2]*

1 HIV Clinic, Trang Hospital, Trang Province, Thailand, 2 Department of Parasitology, Phramongkutklao College of Medicine, Bangkok, Thailand, 3 Department of Microbiology, Faculty of Science, Mahidol University, Bangkok, Thailand, 4 Department of Clinical Immunology, Trang Hospital, Trang Province, Thailand

* ppiyaraj@pcm.ac.th

**Data Availability Statement:** All relevant data are within the manuscript and its Supporting information files.

## Abstract

Leishmaniasis poses a significant health burden, particularly among immunocompromised patients. In Thailand, *Leishmania* infection caused by *Leishmania martiniquensis* and *Leishmania orientalis* lacks information about the incidence and risk factors among HIV-infected populations. This longitudinal cohort study aimed to investigate the incidence and persistence of *Leishmania* infection among HIV-infected individuals in an affected area, Trang Province, Southern Thailand. The study also identified risk factors associated with the incidence of *Leishmania* infection. The study enrolled 373 participants in the HIV clinic, Trang Hospital, who initially tested negative for *Leishmania* infection during 2015–2016, and 133 individuals initially tested positive for *Leishmania* infection. Thus, follow-up visits of 506 participants occurred during 2018–2019. Direct Agglutination Test (DAT) and nested PCR (nPCR) identified incidents and persistent cases of *Leishmania* infection. Cox proportional-hazards regression analyses were performed to assess risk factors for the incidence of *Leishmania* infection. Among the initially negative group, 12 incident cases comprised one *L. orientalis* infection and 11 seropositive cases using DAT, resulting in a cumulative incidence of 3.2% and an incidence density of 10.38 per 1000 person-years. Increasing age was a significant predictor of the incidence of *Leishmania* infection. Five persistent cases comprised one *Leishmania donovani* complex and four seropositive cases using DAT in the initially positive group, with a cumulative persistence rate of 3.7% and a persistence density of 12.85 per 1000 person-years. All patients were asymptomatic. This study sheds light on the incidence and persistence of *Leishmania* infection among HIV-infected individuals in Trang Province, Southern Thailand, underscoring the importance of continued monitoring and tailored interventions to mitigate the impact of this co-infection.

**Funding:** PP was financially supported by Phramongkutklao College of Medicine Research Fund and Mahidol University (Basic Research Fund: fiscal year 2022), and SS was financially supported by Mahidol University (Basic Research Fund: fiscal year 2022). The funders had no role in the study design, data collection and analysis, publication decision, or manuscript preparation.

**Competing interests:** The authors have declared that no competing interests exist.

## Author summary

In Thailand, *Leishmania martiniquensis* and *Leishmania orientalis* are the causative agents of cutaneous (CL) and visceral leishmaniasis (VL) reported in immunocompetent and immunocompromised individuals. VL poses a significant health threat, especially to those with HIV infection. Our study conducted a cohort study among HIV-infected patients in Trang Hospital, Trang Province, Southern Thailand, to explore the incidence and persistence of *Leishmania* infection, a topic not extensively studied in this region. Our findings are crucial for multiple reasons. Firstly, this study reveals a significant incidence rate of 3.2% among HIV patients, and increasing age was a significant predictor of incident *Leishmania* infection. Secondly, the persistence of *Leishmania* infection in 3.7% of initially positive cases underscores the challenges in managing this co-infection. It has immediate implications for healthcare providers, policymakers, and public health officials, aiding them in implementing targeted interventions and preventive measures to reduce the impact of leishmaniasis among HIV-infected individuals. By shedding light on this understudied issue, we hope to contribute to better health outcomes for this vulnerable population.

## Introduction

Leishmaniasis is a neglected tropical disease caused by the protozoan parasite of the genus *Leishmania* that encompasses a broad spectrum of clinical manifestations, including cutaneous (CL), mucocutaneous (ML), and visceral leishmaniasis (VL) [1, 2]. Leishmaniasis is prevalent globally in the Indian subcontinent, East Africa, and South America. Notably, it is endemic in countries such as Sudan, South Sudan, and Ethiopia, where the disease poses significant health challenges [2–4]. The region's environmental and ecological factors and socioeconomic conditions contribute to the persistence and transmission of typical and specific species of *Leishmania* parasites in each region, making it a substantial public health concern [2,4]. Different forms of leishmaniasis are reported in these areas, affecting both immunocompromised and immunocompetent individuals [2,3,5]. It poses a significant health burden, particularly among immunocompromised individuals such as HIV-infected patients [3,6–8]. The co-infection of *Leishmania* and HIV presents unique challenges for diagnosis, treatment, and management, as both diseases can exacerbate each other's clinical manifestations. HIV-infected individuals are at a higher risk of developing severe and disseminated leishmaniasis, increasing morbidity and mortality rates [3,8–11].

In Thailand, leishmaniasis, mainly caused by *Leishmania martiniquensis* and *Leishmania orientalis* of the subgenus *Mundinia*, has been reported in various regions, including Trang Province, Southern Thailand. These two species typically cause CL and VL among immunocompromised and immunocompetent patients [1,5,6,12–16]. Moreover, a typical disseminated CL was also reported in some HIV-infected Thai patients [12].

In our previous studies, the diagnosis of *Leishmania* infection relied on detecting antibodies or DNA. Polymerase chain reaction (PCR) amplifying the internal transcribed spacer 1 region (ITS1) of the SSU-rRNA gene is a commonly used technique to successfully detect *Leishmania* DNA with the sensitivity and specificity of 100% and 82.4%, respectively [17], while antibody detection using the Direct Agglutination Test (DAT), a semi-quantitative serological method, is also helpful to study *Leishmania* infection in Thailand [13] with the estimated sensitivity and specificity of 94.8% and 97.1%, respectively [18]. Unfortunately, the commercial rk39

rapid dipstick test developed from the antigen of *Leishmania donovani* revealed unsuccessful results in detecting the antibodies against VL among infected Thai patients caused by *L. martiniquensis* and *L. orientalis* [19].

Previous studies examining the incidence and risk factors of leishmaniasis caused by *L. donovani* was conducted in Ethiopia [20], Spain [21], India, and Nepal [22]. In India and Nepal, the significance of asymptomatic *L. donovani* infection in the transmission of VL, identifying key risk factors such as age, household members with VL, and geographic differences [22]. In Spain, the study revealed that highly active antiretroviral therapy (HAART) significantly reduced the incidence of symptomatic VL among HIV-1 infected patients, with profound immunosuppression being the leading risk factor [21]. In Ethiopia, VL epidemics were linked to the migration of workers from non-endemic highlands, with higher infection rates observed among male labor migrants during the weeding and harvest seasons [20]. The findings underscore the need for targeted preventive measures and awareness programs to mitigate the risk of leishmaniasis in these diverse settings.

Our previous studies mainly focused on estimating the prevalence of *Leishmania* infection among HIV-infected patients [1 6,13]. However, knowledge of the incidence, persistence, and risk factors of *Leishmania* infection among HIV-infected individuals in Thailand has not been studied. Therefore, a critical need for cohort studies to address this knowledge gap is essential to provide insights into *Leishmania* infection incidence with risk factors among HIV-infected patients, including characteristics of persistent infection. We conducted a longitudinal study among HIV-infected patients in an HIV clinic, Trang Hospital, with baseline enrollment during 2015–2016 and subsequent follow-up visits during 2018–2019 to identify incident and persistent cases. To our knowledge, this is the first cohort study in Thailand to investigate the incidence and persistence of *Leishmania* infection among HIV-infected individuals. Additionally, the outcomes of this study are essential for healthcare providers, policymakers, and public health officials in implementing effective interventions, enhancing surveillance, improving clinical management, and preventive measures to mitigate the impact of leishmaniasis among HIV-infected individuals in the region.

## Methods

### Ethics statement

This study followed the ethical principles outlined in the Declaration of Helsinki. Ethical approval was obtained from the Institutional Review Board of the Royal Thai Army Medical Department (approval number: S034h/60). Before their inclusion in the study, all study participants gave written informed consent. Confidentiality and anonymity of the participants were ensured by assigning unique identification codes to the collected data.

### Study design and setting

This cohort study was conducted at the HIV clinic of Trang Hospital in Trang Province, Southern Thailand. The study enrolled a baseline study population from 2015 to 2016 and conducted follow-up assessments from 2018 to 2019. The Trang Hospital HIV program provided care for approximately 700 HIV-infected patients who were regularly monitored. The clinic adhered to HIV treatment guidelines according to national and WHO recommendations [23]. HAART medication was provided free of charge every three to six months, and CD4 cell count measurements were performed every six months, per national guidelines. During each clinic visit, these patients were screened for opportunistic infections, and if necessary, they received appropriate medical treatment to address any identified health concerns. This comprehensive care approach ensured that HIV-infected individuals received antiretroviral therapy and

required medical attention for other conditions, contributing to their overall health and well-being.

## Participants

The study population consisted of HIV-infected patients receiving medical care at the HIV clinic of Trang Hospital during the baseline enrollment period. Inclusion criteria for participants were as follows: [1] confirmation of HIV infection based on documented medical records, [2] age of 18 years or older, and [3] provision of informed consent to participate in the study.

The study population consisted of 643 HIV-infected individuals, with 479 individuals testing negative for leishmaniasis at baseline, forming the population at risk for incident *Leishmania* infection, and 164 individuals testing positive at baseline, who were further investigated for the persistence of the infection. All participants were asymptomatic and receiving HAART.

## Definition

**Asymptomatic *Leishmania* infection.**   Within the scope of this study, pertains to individuals who had no history of VL infection and did not exhibit any clinical symptoms associated with VL during the study period. However, these individuals tested positive for *Leishmania* infection through diagnostic methods such as the Direct Agglutination Test (DAT) or nested Polymerase Chain Reaction (nPCR) assays.

**Symptomatic visceral leishmaniasis.**   VL is characterized by individuals with a history of fever lasting at least two weeks and splenomegaly. Other clinical manifestations include hepatomegaly, weight loss, anemia, leucopenia, thrombocytopenia, and hypergammaglobulinemia. Microscopic examination or nPCR assay using clinical samples such as bone marrow aspirates, lymph nodes, buffy coat, or other biopsy samples was used to confirm the presence of *Leishmania* parasites [13].

**Seropositivity for *Leishmania* infection.**   Signifies the detection of specific antibodies against the *Leishmania* parasite in individuals. This seropositivity could be detected in individuals whether or not they exhibited symptoms of the infection. Detection of these antibodies indicated that an individual had been exposed to *Leishmania* parasites at some point. DAT was used to assess seropositivity and detect the presence of these antibodies, contributing to diagnosing and monitoring *Leishmania* infection in affected populations.

**The incidence of *Leishmania* infection.**   Pertains to newly diagnosed cases among HIV-infected patients in Trang Hospital, specifically during the study's follow-up period, which spanned from 2018 to 2019. These cases involved individuals who initially tested negative for *Leishmania* infection at the baseline enrollment phase conducted between 2015 and 2016 but subsequently positive for *Leishmania* infection using nPCR or DAT (seroconversion).

**The persistence of *Leishmania* infection.**   Pertains to cases of the disease among HIV-infected patients who initially tested positive for leishmaniasis during the baseline enrollment between 2015 and 2016 and detectable *Leishmania* infection during the follow-up period from 2018 to 2019, where positivity was determined using nPCR and DAT.

## Data collection

**Baseline enrollment.**   Between 2015 and 2016, we gathered baseline information from enrolled participants using structured questionnaires and a review of their medical records. This comprehensive dataset encompassed socio-demographic details, HIV-related parameters including CD4 count, HIV viral load, antiretroviral therapy regimen, and treatment duration, and potential risk factors linked to *Leishmania* infection, such as travel history to

regions where the disease was prevalent, occupation, and exposure to habitats inhabited by the vectors.

All eligible participants underwent a thorough screening for *Leishmania* infection during the initial enrollment phase. Individuals who tested negative for *Leishmania* infection at this baseline assessment were included in the subsequent evaluation to determine the incidence of *Leishmania* infection. Meanwhile, those who tested positive during this initial screening were eligible for further evaluation to study the persistence of the infection.

### Follow-up visit

From 2018 to 2019, we conducted a follow-up study to investigate the incidence and persistence of *Leishmania* infection among participants. We carried out active surveillance during this period, which involved ongoing communication with participants and scheduled clinic visits. Participants were also educated about the signs and symptoms of leishmaniasis and encouraged to seek medical attention if they experienced any suggestive symptoms.

All participants underwent thorough evaluation throughout the follow-up period, including clinical examinations and laboratory tests such as DAT and nPCR. Based on established diagnostic criteria, these diagnostic measures were employed to confirm or rule out *Leishmania* infection. This comprehensive approach allowed us to closely monitor the occurrence and persistence of *Leishmania* infection among the study participants during the specified timeframe.

### Samples collection and preparation

Collecting and preparing samples for the research on incident and persistent *Leishmania* infection among HIV-infected patients in Trang Province, Southern Thailand, involved the collection of blood samples from each participant [13]. A total volume of 8 mL was obtained from each patient using an ethylenediaminetetraacetic acid (EDTA) anticoagulant tube for the blood samples. The plasma and the buffy coat were collected and then preserved at -20˚C to maintain stability and prevent degradation until further analysis [13].

### Detection of *Leishmania* antibodies

The DAT, a semi-quantitative test, was used to detect *Leishmania* antibodies. DATkit from KIT Biomedical Research in Amsterdam, the Netherlands, was utilized, following the manufacturer's instructions. The DAT contained the freeze-dried *Leishmania* antigen prepared from the axenic culture of *Leishmania donovani* promastigotes. The plasma samples obtained from the study participants were used to perform the DAT. A positive control was included in the test, consisting of plasma samples from individuals who had confirmed VL cases. These samples were previously verified using the nPCR method to ensure their accuracy as positive controls. On the other hand, a negative control was also included in the test, which comprised plasma samples from healthy individuals who did not have *Leishmania* infection. Two experienced technicians interpreted the DAT results based on titers, representing the dilution of the plasma samples that still resulted in agglutination. A titer of ≥1:100 was considered a positive result for *Leishmania* antibodies, as recommended by the manufacturer of the DATkit [13]. Using the DAT to detect *Leishmania* antibodies in the plasma samples, those exposed to the parasite developed an immune response in the form of specific antibodies that could be detected.

## Detection of *Leishmania* DNA

*Leishmania* DNA was detected using samples of buffy coat collected from each participant. Briefly, 200 ml of the buffy coat was taken from each participant. These samples were then subjected to the DNA extraction process using the Gen UPTM gDNA Kit [13]. The extracted DNA was eluted in a final volume of 40 μL and stored at a temperature of -20˚C. The nPCR assay targeted the ITS1 region of SSU-rRNA of *Leishmania* and amplified the *Leishmania* DNA precisely [13].

## Sequence analysis

After the nPCR assay, the positive PCR products were sent to U2Bio Co. Ltd. in Seoul, South Korea, for sequence analysis. The chromatograms of the obtained sequences were carefully validated to ensure the accuracy and reliability of the sequencing results using BioEdit version 7.0.1, a software tool from Ibis Therapeutics in Carlsbad, CA. Once validated, the sequences were aligned with reference *Leishmania* strains from the GenBank database to accurately identify the specific species of *Leishmania* present in the samples.

## Data analysis

Descriptive statistics were used to summarize the baseline characteristics of the study population using the chi-square test. Incidence density of *Leishmania* infection was calculated as the number of individuals who newly reported *Leishmania* infection divided by the total number of person-years of follow-up among HIV-infected patients who were negative for *Leishmania* infection at baseline. Person-time was calculated as the midpoint between the date of the last report of negative *Leishmania* infection and the date of the first report of *Leishmania* infection, or, for those consistently reporting no *Leishmania* infection, the date of the last visit. Persistence of *Leishmania* infection was determined by calculating the incidence density of persistent *Leishmania* infection as the number of individuals with persistent *Leishmania* infection divided by the total number of person-years of follow-up among those who tested positive for *Leishmania* infection at baseline. The date of persistent *Leishmania* infection was defined as the midpoint between the date of the last positive *Leishmania* test and the first positive *Leishmania* test or the date of the last visit for those consistently positive for *Leishmania* infection.

Cox proportional hazards regression analysis was used to identify risk factors associated with developing *Leishmania* infection, including age, sex, CD4 count, HIV viral load, and potential risk factors identified in the baseline assessment. Participants with missing data or records due to missed follow-up visits were excluded from the analysis. The proportional hazards assumption for Cox regression analysis was assessed using scaled Schoenfeld residuals. Hazard ratios (HRs) and 95% confidence intervals (CIs) were calculated for the associations, and P-values of 0.05 or less were considered statistically significant. All statistical analyses were conducted using STATA, version SE18.

# Results

## Characteristics of the study population

Fig 1 presents an overview of the study's baseline data from 2016, detailing the enrollment of 643 HIV-infected individuals, their *Leishmania* infection test results, and the subsequent follow-up outcomes in 2019. At baseline, 164 individuals tested positive for *Leishmania* infection, with 109 being DAT positive only, 44 nPCR positive only, and 11 positives by both DAT and nPCR, while 479 individuals tested negative. During the follow-up period in 2019, 373 of the 479 individuals who were negative at baseline (77.9%) were followed up, with 12 (3.2%) testing

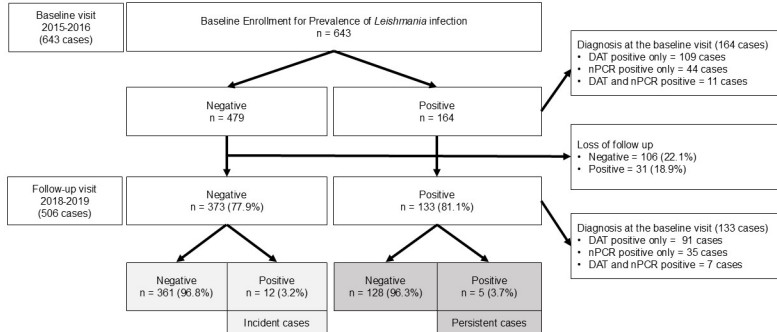

**Fig 1. Population flow chart showing participant recruitment and follow-up.**

positive for *Leishmania* infection, indicating incident cases. Among the 164 individuals who were positive at baseline, 133 (81.1%) were followed up; of these, five individuals (3.7%) remained positive, indicating persistent infection, with 91 initially DAT positive only, 35 nPCR positive only, and seven positives by both DAT and nPCR. Meanwhile, 128 individuals (96.3%) who were positive at baseline tested negative during follow-up, indicating clearance of the infection. Loss to follow-up was noted in 106 (22.1%) of the individuals who were negative at baseline and 31 (18.9%) of those who were positive at baseline. The median follow-up time was 2.9 years (IQR: 2.5–3.4 years).

Table 1 provides a comprehensive overview of the study population's characteristics, categorized by their *Leishmania* infection status at baseline (2015–2016). All participants were asymptomatic, had no history of VL, and received HAART.

## The population at risk for incident *Leishmania* infection (tested negative at baseline, n = 479)

Table 1 shows the initial cohort; 373 cases were successfully followed. Among those who enrolled in the follow-up visit, the characteristics were diverse. The mean age of all participants is 44.1 years, with a standard deviation of 8.2, with an age range of 21.6 to 71.3 years. The total number of male participants is 189 (50.7%), and the total number of female participants is 184 (49.3%). Education levels varied, with 43.4% having completed primary school or lower, 30.1% having secondary education, 12.3% having vocational school qualifications, and 14.2% holding a bachelor's degree or higher. Employment also exhibited diversity, including 8.3% unemployed individuals, 30.8% engaged in agriculture, 7.5% in government jobs, 20.9% in business, and 23.3% as laborers.

## The population at risk for persistent *Leishmania* infection (tested positive at baseline, n = 164)

This group encompasses individuals who initially tested positive for *Leishmania* infection at baseline and were subsequently followed up to investigate the persistence of the infection during the follow-up period (2018–2019). We successfully followed 133 (81.1%) in this group, while 31 (18.9%) lost to follow-up. Those who enrolled in the follow-up visit exhibited a range of characteristics. The mean age of all participants is 43.0 years, with a standard deviation of 12.0 and an age range of 20.3 to 73.7 years. Males constitute 46.6% of the participants.

Education levels varied, with 42.1% having completed primary school or lower, 25.6% having secondary education, 18.0% having vocational school qualifications, and 14.3% holding a

**Table 1. Characteristics of the study population by *Leishmania* infection status at baseline.**

| Characteristics | *Leishmania* infection among negative cases at baseline, n = 373 | | | | *Leishmania* infection among positive cases at baseline, n = 133 | | | |
|---|---|---|---|---|---|---|---|---|
| | Negative | Positive | Total | P-value | Negative | Positive | Total | P-value |
| | n = 361 (%) | n = 12 (%) | n = 373 (%) | | n = 128 (%) | n = 5 (%) | n = 133 (%) | |
| **Age (year)** | | | | 0.805 | | | | 0.756 |
| Mean (SD) | 43.9 (8.1) | 47.4 (10.8) | 44.1 (8.2) | | 43.1 (8.8) | 44.9 (12.0) | 43 (8.9) | |
| Min-Max | 21.6–70.0 | 36.7–71.3 | 21.6–71.3 | | 20.3–73.7 | 29.1–62.8 | 20.3–73.7 | |
| **Sex** | | | | 0.962 | | | | 0.224 |
| Male | 183 (50.7) | 6 (50.0) | 189 (50.7) | | 61 (47.7) | 1 (20.0) | 62 (46.6) | |
| Female | 178 (49.3) | 6 (50.0) | 184 (49.3) | | 67 (52.3) | 4 (80.0) | 71 (53.4) | |
| **Education** | | | | 9.966 | | | | 0.254 |
| Primary school and lower | 157 (43.5) | 5 (41.7) | 162 (43.4) | | 55 (42.9) | 1 (20.0) | 56 (42.1) | |
| Secondary school | 108 (29.9) | 4 (33.3) | 112 (30.0) | | 31 (24.2) | 3 (60.0) | 34 (25.6) | |
| Vocational school | 45 (12.5) | 1 (8.3) | 46 (12.3) | | 24 (18.7) | 0 | 24 (18.0) | |
| Bachelor or higher | 51 (14.1) | 2 (16.7) | 53 (14.2) | | 18 (14.1) | 1 (20.0) | 19 (14.3) | |
| **Occupation** | | | | 0.135 | | | | 0.461 |
| Unemployed | 31 (8.6) | 0 | 31 (8.3) | | 13 (10.2) | 0 | 13 (9.8) | |
| Agriculture | 114 (31.6) | 1 (8.3) | 115 (30.8) | | 35 (27.3) | 3 (60.0) | 38 (28.6) | |
| Government | 27 (7.5) | 1 (8.3) | 28 (7.5) | | 11 (8.6) | 0 | 11 (8.3) | |
| Business | 76 (21.0) | 2 (16.7) | 78 (20.9) | | 27 (21.1) | 1 (20.0) | 28 (21.0) | |
| Laborer | 82 (22.7) | 5 (41.7) | 87 (23.3) | | 31 (24.2) | 0 | 31 (23.3) | |
| Other | 31 (8.6) | 3 (25.0) | 34 (9.1) | | 11 (8.6) | 1 (20.0) | 12 (9.0) | |
| **Intravenous drug user** | | | | 0.135 | | | | 0.531 |
| No | 304 (84.2) | 12 (100) | 316 (84.7) | | 114 (89.1) | 4 (80.0) | 118 (88.7) | |
| Yes | 57 (15.8) | 0 | 57 (15.3) | | 14 (10.9) | 1 (20.0) | 15 (11.3) | |
| **Recreational drug user** | | | | 0.618 | | | | 0.338 |
| No | 316 (87.5) | 12 (100) | 328 (87.9) | | 108 (84.4) | 5 (100) | 113 (84.9) | |
| Yes | 45 (12.5) | 0 | 45 (12.1) | | 20 (15.6) | 0 | 20 (15.1) | |
| **Travel abroad** | | | | 0.569 | | | | 0.207 |
| No | 310 (85.9) | 11 (91.7) | 321 (86.1) | | 115 (89.8) | 5 (100) | 120 (90.2) | |
| Yes | 51 (14.1) | 1 (8.3) | 52 (13.9) | | 13 (10.2) | 0 | 13 (9.8) | |
| **Stilt house** | | | | 0.393 | | | | 0.771 |
| No | 304 (84.2) | 9 (75.0) | 313 (83.9) | | 95 (74.2) | 4 (80.0) | 99 (74.4) | |
| Yes | 57 (15.8) | 3 (25.0) | 60 (16.1) | | 33 (25.8) | 1 (20.0) | 34 (25.6) | |
| **Raised animal** | | | | 0.716 | | | | 0.539 |
| No | 340 (94.2) | 11 (91.7) | 351 (94.1) | | 119 (92.9) | 5 (100) | 124 (93.2) | |
| Yes | 21 (5.8) | 1 (8.3) | 22 (5.9) | | 9 (7.1) | 0 | 9 (6.8) | |
| **Bed net** | | | | 0.644 | | | | 0.409 |
| Yes | 127 (35.2) | 5 (41.7) | 132 (35.4) | | 53 (41.4) | 3 (60.0) | 56 (42.1) | |
| No | 234 (64.8) | 7 (58.3) | 241 (64.6) | | 75 (58.6) | 2 (40.0) | 77 (57.9) | |
| **Opportunistic infection** | | | | 0.604 | | | | 0.667 |
| No | 343 (95.0) | 11 (91.7) | 354 (94.9) | | 111 (86.7) | 4 (80.0) | 115 (86.5) | |
| Yes | 18 (4.9) | 1 (8.3) | 19 (5.1) | | 17 (13.3) | 1 (20.0) | 18 (13.5) | |
| **Current Viral load** | | | | 0.899 | | | | 0.530 |
| Undetectable | 327 (90.6) | 11 (91.7) | 338 (90.6) | | 114 (89.1) | 4 (80.0) | 118 (88.7) | |
| Detectable | 34 (9.4) | 1 (8.3) | 35 (9.4) | | 14 (10.9) | 1 (20.0) | 15 (11.3) | |
| **Current CD4 cell count** | | | | 0.832 | | | | 0.458 |
| >500 cell/mm$^3$ | 172 (47.6) | 7 (58.3) | 179 (47.9) | | 45 (35.2) | 3 (60.0) | 48 (36.1) | |
| 201–500 cell/mm3 | 157 (43.4) | 4 (33.3) | 161 (43.1) | | 69 (53.9) | 1 (20.0) | 70 (53.7) | |
| < = 200 cell/mm$^3$ | 32 (8.9) | 1 (8.3) | 33 (8.8) | | 14 (10.9) | 1 (20.0) | 15 (11.3) | |

bachelor's degree or higher. Employment also showed diversity, including 9.8% unemployed individuals, 28.6% engaged in agriculture, 8.3% in government jobs, 21.0% in business, and 23.3% as laborers. Importantly, these characteristics did not display statistically significant differences between those retained and those lost to follow-up (S1 Table), except for CD4 cell counts, which was not statistically significant (P = 0.302), revealed that a higher proportion of individuals with persistent *Leishmania* infection had lower CD4 cell counts among those who enrolled in the follow-up visit. Nonetheless, the overall pattern of characteristics within the enrolled participants suggested their representativeness of the initial population.

S1 Table categorized by *Leishmania* infection status at baseline and comparing characteristics among those who followed up and those who were lost to follow-up using the chi-square test, indicated that the retained participants closely resemble the overall population; this strengthened the reliability of the study's findings concerning incident and persistent *Leishmania* infections among HIV-infected patients. Most characteristics did not demonstrate statistically significant differences between those retained and those lost to follow-up. Only the intravenous drug use was a statistically significant difference between those who were follow-up and those lost to follow-up groups.

## Incidence of *Leishmania* infection

During the follow-up period (2018–2019), among 373 individuals who initially tested negative for *Leishmania* infection at baseline in 2015–2016, 12 cases tested positive for *Leishmania* infection. These 12 incident cases contributed to a total of 1155.4 person-years of follow-up. The cumulative incidence of *Leishmania* infection was 3.2%, indicating that approximately 3.2% of individuals initially testing negative at baseline developed *Leishmania* infection during the follow-up period. Additionally, the incidence density, representing the rate of new infections per 1000 person-years, was computed as 10.38 per 1000 person-years, providing a more granular measure of the infection rate within this cohort.

Among the positive cases, 11 individuals tested positive for *Leishmania* antibodies using DAT alone. The distribution of seropositive cases varied across different titers. Specifically, two cases were at a titer of 1:100, followed by six at 1:200, and three at 1:400. Notably, one individual tested positive for *Leishmania* DNA in the buffy coat sample using nPCR, and species identification revealed *L. orientalis*. Within the study period, all 12 incident cases were asymptomatic.

## Persistence of *Leishmania* infection

The persistence of *Leishmania* infection pertains to cases among HIV-infected patients who initially tested positive for leishmaniasis during the baseline enrollment between 2015 and 2016, where positivity was determined using both the Direct Agglutination Test (DAT) and nPCR. At follow-up between 2018 and 2019, among the 133 individuals who were followed up after initially testing positive, five individuals (3.7%) remained positive, indicating persistent infection. Regarding the methodology used to assess persistence, both DAT and nPCR were utilized during the follow-up period. Specifically, all 133 individuals who tested positive at baseline were retested using both DAT and nPCR during follow-up.

Table 2 shows the results of DAT and species identification among 133 *Leishmania* infections at the baseline visit (2015–2016). During the follow-up period (2018–2019), five cases remained identified as testing positive for *Leishmania* infections (Table 3). These five persistent cases contributed to a total of 389 person-years of follow-up. The cumulative persistence of *Leishmania* infection was 3.7%, indicating that approximately 3.7% of individuals initially testing positive at baseline remained infected with *Leishmania* infections during the follow-

**Table 2. Results of DAT and species identification among 133 *Leishmania* infections at the baseline visit (2015–2016).**

| | Diagnosis of *Leishmania* infection at Baseline visit (2015–2016) and enrolled at the follow-up visit (2018–2019) n = 133 | | | |
|---|---|---|---|---|
| | **Positive by DAT only (n = 91)** | **Positive by nPCR (n = 35)** | **Positive by DAT and nPCR (n = 7)** | **Total (n = 133)** |
| DAT 1:100 | 6 | | | |
| DAT 1:200 | 12 | | | |
| DAT 1:400 | 38 | | *L. martiniquensis* (1), *Leishmania.* spp. (2) | |
| DAT 1:800 | 26 | | *L. orientalis* (3) | |
| DAT 1:1600 | 9 | | *L. orientalis* (1) | |
| **Species identification** | | | | |
| *L. orientalis* | | 13 | | |
| *L. donovani complex* | | 7 | | |
| *L. martiniquensis* | | 5 | | |
| *L. lainsoni* | | 4 | | |
| *Leishmania* spp. | | 6 | | |
| | 91 | 35 | 7 | 133 |

**Table 3. Results of DAT and nPCR among five persistent cases at the follow-up visit (2018–2019).**

| Patient number | Diagnosis at the baseline visit | Diagnosis at the follow-up visit |
|---|---|---|
| 1 | DAT 1:200 | DAT 1:100 |
| 2 | DAT 1:400 | DAT 1:100 |
| 3 | DAT 1:200 | DAT 1:200 |
| 4 | *L. lainsoni* | DAT 1:400; Negative nPCR |
| 5 | *Leishmania* spp. | *L. donovani* complex |

up period. Additionally, the persistence density, representing the rate of persistent infections per 1000 person-years, was 12.85 per 1000 person-years, providing a more detailed measure of the persistence rate within this cohort. However, all of the persistent cases were asymptomatic.

Table 3 shows four individuals who tested positive for *Leishmania* antibodies using DAT alone among the five persistent cases. The distribution of seropositive cases varied across different titers, with two cases observed at a titer of 1:100, one at 1:200, and one at 1:400. Additionally, one individual tested positive for *Leishmania* DNA in their buffy coat samples using nPCR, and species identification revealed *L. donovani* complex.

## Risk factors for incident *Leishmania* infection

We conducted bivariate and multivariate Cox proportional-hazards regression analyses to identify risk factors associated with incident *Leishmania* infection among the study population, as shown in Table 4. Age was treated as a continuous variable in the analysis. In the bivariate analysis, the hazard ratio (HR) for age was 1.05 (95% CI: 0.99–1.12, P = 0.103). This association was statistically significant using the multivariate analysis, with an HR of 1.09 (95% CI: 1.02–1.17, P = 0.013), indicating that increasing age was a significant predictor of incident *Leishmania* infection. Sex, education levels, and travel history were not significant risk factors for incident *Leishmania* infection in both bivariate and multivariate analyses. Similarly, occupation, intravenous drug use, recreational drug use, housing conditions (stilt house and raised animals), bed net use, opportunistic infections, viral load, and CD4 cell counts were not significantly associated with the risk of incident *Leishmania* infection.

**Table 4. Incident *Leishmania* infection, bivariate and multivariate Cox proportional-hazards regression analyses of risk factors for incident *Leishmania* infection (n = 373 (1155 person-year of follow-up), negative *Leishmania* infection at baseline; n = 12, *Leishmania* infection at follow-up).**

| Characteristics | Incident *Leishmania* infection (cases/person-year) | Incidence density *Leishmania* infection /1000 Person-Years (95% CI) | Bivariate analysis hazard ratio (95% CI) | P-value | Multivariate analysis hazard ratio (95% CI) | P-value |
|---|---|---|---|---|---|---|
| **Age (year)** | 12/1155 | 10.3 (5.8–18.3) | 1.05 (0.99–1.12) | 0.103 | 1.09 (1.02–1.17) | 0.013 |
| **Sex** | | | | | | |
| Male | 6/581.3 | 10.3 (4.6–22.9) | 1.00 | | 1.00 | |
| Female | 6/574.1 | 10.4 (4.7–23.3) | 0.92 (0.29–2.86) | 0.885 | 0.95 (0.26–3.41) | 0.938 |
| **Education** | | | | | | |
| Primary school and lower | 5/509.8 | 9.8 (4.1–23.6) | 1.00 | | 1.00 | |
| Secondary school | 4/346.7 | 11.5 (4.3–30.7) | 1.29 (0.35–4.82) | 0.702 | 3.04 (0.65–14.26) | 0.159 |
| Vocational school | 1/136.9 | 7.3 (1.0–51.8) | 1.31 (0.15–11.4) | 0.809 | 2.91 (0.28–30.07) | 0.369 |
| Bachelor or higher | 2/161.9 | 12.3 (3.1–49.4) | 1.31 (0.25–6.74) | 0.749 | 1.98 (0.32–12.12) | 0.448 |
| **Travel abroad** | | | | | | |
| No | 11/995.3 | 11 (6.1–19.9) | 1.00 | | 1.00 | |
| Yes | 1/160.1 | 6.2 (0.9–44.3) | 0.56 (0.07–4.31) | 0.575 | 0.40 (0.05–3.32) | 0.400 |
| **Stilt house** | | | | | | |
| No | 9/969.3 | 9.3 (4.8–17.8) | 1.00 | | 1.00 | |
| Yes | 3/186.1 | 16.1 (5.2–49.9) | 1.73 (0.47–6.39) | 0.412 | 2.51 (0.58–10.82) | 0.218 |
| **Raised animal** | | | | | | |
| No | 11/1088.2 | 10.1 (5.5–18.2) | 1.00 | | 1.00 | |
| Yes | 1/67.2 | 14.9 (2.1–10.5) | 1.2 (0.15–9.30) | 0.861 | 1.49 (0.17–13.15) | 0.714 |
| **Bed net use** | | | | | | |
| Yes | 5/411.8 | 12.1 (5.0–29.2) | 1.00 | | 1.00 | |
| No | 7/743.6 | 9.4 (4.5–19.7) | 0.87 (0.28–2.77) | 0.819 | 0.48 (0.13–1.78) | 0.276 |
| **Opportunistic infection** | | | | | | |
| No | 11/1093 | 10.1 (5.6–18.2) | 1.00 | | 1.00 | |
| Yes | 1/62.4 | 16 (2.2–11.4) | 1.29 (0.17–10.1) | 0.804 | 1.78 (0.18–17.18) | 0.619 |
| **Current Viral load** | | | | | | |
| Undetectable | 11/1043.5 | 10.5 (5.8–19.0) | 1.00 | | 1.00 | |
| Detectable | 1/111.9 | 8.9 (1.2–6.3) | 0.65 (0.08–5.21) | 0.687 | 0.84 (0.06–11.91) | 0.900 |
| **Current CD4 cell count (cell/mm3)** | | | | | | |
| >500 | 10/848.7 | 11.8 (6.3–21.9) | 1.00 | | 1.00 | |
| 201–500 | 1/201.4 | 4.9 (0.7–35.2) | 0.39 (0.05–3.11) | 0.380 | 0.37 (0.04–3.35) | 0.377 |
| ≤200 cell/mm$^3$ | 1/105.2 | 9.5 (1.3–67.4) | 0.66 (0.08–5.24) | 0.691 | 1.05 (0.08–13.11) | 0.968 |

## Discussion

Our study represents a pioneering effort in Thailand, providing the first longitudinal investigation into the incidence and persistence of *Leishmania* infection among HIV-infected individuals. This research also marks the initial report of the risk factors associated with the incidence of *Leishmania* infection in this unique patient population.

A study of asymptomatic *Leishmania* infection among patients with HIV in an urban area of Brazil using different diagnostic tests, ELISA, IFTA, DAT, and PCR, showed weak concordance of the tests utilized [24]. Thus, the selection of diagnostic tests must be considered for each study. This study used DAT to detect antibodies against *Leishmania* infection since the test was simple and rapid and revealed high sensitivity and specificity. One of the limitations

of DAT is that it can detect cross-reaction, especially with *Trypanosoma cruzi* infection [25]. The animal *Trypanosoma* group is enzootic, and to date, human Trypanosomiasis has not been reported in Thailand. Thus, DAT is still helpful in diagnosing *Leishmania* infection in Thailand.

Identifying 12 incident cases of *Leishmania* infection that comprised 11 seropositive cases and one *L. orientalis* infection by nPCR among individuals who initially tested negative for the parasite at baseline is a crucial finding that indicates an ongoing risk of *L. orientalis* transmission in the affected area of Trang Province. Throughout the follow-up period, these 12 individuals contributed 1155.4 person-years of follow-up. The cumulative incidence of 3.1% observed in this study underlines the vulnerability of individuals with HIV to *Leishmania* infection in this region. This rate of incidence, representing approximately 31 new infections per 1000 person-years, highlights the substantial burden of *Leishmania* infection among individuals with HIV in Thailand. The number underscores the significant burden of *Leishmania* infection of HIV-infected patients in Thailand.

Compared to other settings, such as North West Ethiopia [4], where our findings fall within the range of 3.2–9.5%, our incidence rate in Thailand appears relatively lower but consistent with the epidemiological profile of *Leishmania* infection among HIV patients in this region. However, a study in Western Tigray, Ethiopia, reported a notably higher incidence rate of 8.4% [20], possibly due to the virulence of species of *Leishmania*, regional environmental factors, vector density, or variations in healthcare infrastructure. In India and Nepal, high-endemic areas of VL caused by *L. donovani* revealed incident asymptomatic *L. donovani* infection among immunocompetent individuals nine times more frequent than incident VL disease [22]. Ostyn et al. (2011) reported that about one in 50 new incident cases, but latent infections, developed VL within the next 18 months, while more than 80% became seronegative within one year [26]. Additionally, Das et al. (2020) studied the conversion of asymptomatic infection to symptomatic *L. donovani* infections among immunocompetent individuals in highly endemic areas in India; 23.80% of asymptomatic converted into the disease were reported [27].

In contrast, China exhibited a much lower incidence rate of 3.8 per 100,000 among HIV patients [28], likely influenced by differences in *Leishmania* species, sandfly vectors, or prevention and control measures. These comparisons emphasize the importance of considering local epidemiological factors and healthcare practices when assessing *Leishmania* infection risk among individuals with HIV. Regional variations in *Leishmania* species, vector habitats, and preventive strategies can all contribute to differences in incidence rates. Our findings in Thailand underscore the ongoing risk of *Leishmania* transmission among those initially testing negative, highlighting the need for continued vigilance and tailored interventions in this region.

In addition to identifying incident cases, these individuals were closely monitored over a cumulative follow-up period of 389 person-years, resulting in a persistence rate of 3.7%. In this study, the persistence density of 12.85 cases per 1000 person-years emphasized the need for more effective control and follow-up strategies and an understanding of the immune responses in controlling *Leishmania* infection among HIV-infected patients. A study in Western Sicily, Italy, showed that antibodies against *Leishmania* infection could persist for many years and decline slowly but steadily [18]. Notably, the persistence of these specific antibodies was not necessarily linked to a poor therapeutic response or prognosis. However, a sudden increase in antibody levels might be a sentinel sign of a visceral leishmaniasis relapse. In contrast, the sustained presence of high antibody levels could suggest resistance to therapy. Thus, further investigation and understanding of the dynamics of *Leishmania* infection and its persistence in different populations and regions is needed.

Our study highlighted five cases of persistent *Leishmania* infection among individuals who initially tested positive at baseline. Of five cases, four persistent cases revealed low titers of DAT (1:100–1:400) at baseline visits in 2015–2016, signifying the presence of antibodies against the parasite. Only one persistent *L. donovani complex* infection was detected at a follow-up visit in 2018–2019. The previous infections were caused by *L. orientalis*, *L. martiniquensis*, *L. donovani* complex, *L. lainsoni*, and *Leishmania* spp. at the baseline visits 2015–2016, turned to negative nPCR at follow-up visits 2018–2019. In this circumstance, intact immune status by effective use of highly active antiretroviral therapy (HAART) affected *Leishmania*-infected individuals' susceptibility to progress to VL.

The potential for reinfection remains a significant concern, particularly in endemic areas. Reinfection could complicate the clinical management and long-term outcomes for HIV-infected individuals. Therefore, ongoing monitoring and timely interventions are crucial to prevent reinfection and manage persistent cases effectively. Further research is needed to understand the dynamics of *Leishmania* infection and its persistence across different populations and regions, which will be essential in developing targeted strategies to combat this neglected tropical disease.

In this study, *L. lainsoni* infection was identified using the ITS1-PCR; this species was initially reported in Latin America [29]. This patient had no history of traveling abroad. Unfortunately, the effort to confirm *L. lainsoni* infection using other target genes, i.e., *hsp70*-PCR and *kDNA* PCR, was unsuccessful and negative PCR results were obtained. Therefore *L. lainsoni* is still a doubtful species identification.

Further research employing quantitative polymerase chain reaction (qPCR) [30], loop-mediated isothermal amplification (LAMP) [31], and next-generation sequencing (NGS) [32] should be performed since these methods demonstrated high sensitivity and specificity in detecting *Leishmania* infections among HIV-infected individuals. NGS has provided detailed insights into species-specific responses and treatment outcomes by differentiating *Leishmania* species in co-infected individuals. These techniques enhance detection accuracy and deepen our understanding of the molecular interactions between HIV and *Leishmania*. Integrating one of these methodologies into our study emphasizes the need for cutting-edge diagnostic tools in endemic regions to improve the management and control of *Leishmania*-HIV co-infections.

In this study, all incidents and persistent cases were asymptomatic, with no history of symptomatic VL. This asymptomatic nature of the cases is particularly noteworthy. It suggests *Leishmania* infection in individuals with HIV may not always manifest with clinical symptoms, making diagnosis and management even more challenging [4,26,33,34]. The reliance on serological tests, such as DAT, to identify most cases underscores the importance of routine screening and surveillance in HIV care settings. It also emphasizes the need for heightened awareness among healthcare providers regarding the potential for asymptomatic *Leishmania* infection in HIV-infected patients. Continuous monitoring, early detection, and tailored interventions are necessary to combat *Leishmania* infection within the context of HIV effectively.

Our study also identified older age as a significant risk factor for the incidence of *Leishmania* infection among HIV-infected patients. This finding aligns with previous research conducted in various geographical regions. Older individuals may face an increased risk due to age-related immune system changes. Thus, age-specific preventive strategies and heightened clinical monitoring are needed in older HIV-infected patients. We discern similarities and differences by comparing our findings with prior research [35,36]. The observed association between older age and an increased risk of *Leishmania* infection mirrors results from other studies [35–38]. However, the significance and magnitude of this association may vary depending on the study population and geographical context. Studying this association among

the Thai people filled a gap in the literature and provided valuable insights into *Leishmania* infection's epidemiology in Southeast Asia.

In contrast to earlier investigations, our analysis did not reveal significant associations between *Leishmania* infection and male gender, education level, occupation, substance use, housing conditions, bed net use, or CD4 cell count [4,20,21,26,33,34,39]. These differences may arise from the distinct characteristics of our study population or the particular epidemiological context of Thailand. *Leishmania* infection risk factors are multifaceted, influenced by geographical location, local *Leishmania* species, and the HIV epidemic's stage in a given area. Thus, our findings underscored the complexity of these risk factors and the importance of designing interventions for different populations' specific needs. One noteworthy aspect of our study was the absence of an association between *Leishmania* infection and CD4 cell count, which diverges from prior research, suggesting that individuals with lower CD4 counts were at a higher risk of *Leishmania* infection [13,40]. Our study's lack of this association could be attributed to the higher CD4 counts in our HIV-infected population or the effective use of HAART that enhanced immune responses, potentially mitigating *Leishmania* infection risk in HIV-infected individuals [21,41].

This study encountered some limitations. Firstly, the study's single-hospital setting limited the generalizability of the findings to other regions or populations. Hospital-based studies inherently introduced selection bias, primarily capturing individuals seeking care within that facility. Consequently, this study's incidence and risk factors could not fully represent Thailand's broader population of HIV-infected individuals. Thus, future research should aim for multicenter, community-based studies to comprehensively understand *Leishmania* infection in diverse settings. Secondly, the reliance on self-reported data for certain risk factors, such as travel history and drug use, could introduce recall bias. Future studies could incorporate more objective measures or corroborate self-reported information through additional data sources or medical records to address this limitation.

In conclusion, this study significantly enhances our understanding of the dynamics of *Leishmania* infection among HIV-infected individuals in a Southern province of Thailand by identifying key risk factors, incidence, and persistence of the infection and highlighting the predominantly asymptomatic nature of cases. Identifying increased age as a significant risk factor underscores the need for age-specific preventive measures and clinical monitoring. The findings suggest the importance of integrating routine screening for *Leishmania* infection in HIV care programs to enable early detection. Furthermore, the data support the development of public health initiatives focused on improving awareness, training healthcare workers, and enhancing community education on the risks and symptoms of *Leishmania* infection. Future research should aim for greater diversity in study settings and improved data collection methods, as well as prioritize the development of rapid, simplified diagnostic tools for local use. Addressing *Leishmania* infection among HIV-infected individuals is crucial for improving their overall health outcomes and reducing the burden of this neglected tropical disease in endemic regions, thereby advancing public health efforts.

## Supporting information

**S1 Table. Characteristics of the study population by *Leishmania* infection status at baseline (2015–2016) and follow-up visit status (2018–2019).**
(DOCX)

**S1 Data. Minimal data set.**
(XLSX)

## Acknowledgments

We thank all those who contributed to the completion of this study, especially the participants, for their willingness to be part of this research.

## Author Contributions

**Conceptualization:** Lertwut Bualert, Mathirut Mungthin, Saovanee Leelayoova, Phunlerd Piyaraj.

**Formal analysis:** Phunlerd Piyaraj.

**Funding acquisition:** Mathirut Mungthin, Saovanee Leelayoova, Phunlerd Piyaraj.

**Investigation:** Lertwut Bualert, Toon Ruang-areerate, Mathirut Mungthin, Saovanee Leelayoova, Suradej Siripattanapipong, Tawee Naaglor, Nattapong Hongsimakul, Supicha Sroythong, Phakhajee Rattanalertpaiboon, Preeyaporn Tulpeng, Phunlerd Piyaraj.

**Methodology:** Lertwut Bualert, Toon Ruang-areerate, Mathirut Mungthin, Saovanee Leelayoova, Phunlerd Piyaraj.

**Project administration:** Phunlerd Piyaraj.

**Supervision:** Mathirut Mungthin, Saovanee Leelayoova.

**Writing – original draft:** Phunlerd Piyaraj.

**Writing – review & editing:** Mathirut Mungthin, Saovanee Leelayoova.

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
