## [Decision Letter · Decision Letter 0]

6 May 2024

Dear Assoc. Prof. Piyaraj,

Thank you very much for submitting your manuscript "Incidence and persistence of asymptomatic Leishmania infection among HIV-infected patients in Trang province, southern Thailand: a cohort study" for consideration at PLOS Neglected Tropical Diseases. As with all papers reviewed by the journal, your manuscript was reviewed by members of the editorial board and by several independent reviewers. In light of the reviews (below this email), we would like to invite the resubmission of a significantly-revised version that takes into account the reviewers' comments. 

We cannot make any decision about publication until we have seen the revised manuscript and your response to the reviewers' comments. Your revised manuscript is also likely to be sent to reviewers for further evaluation.

Sincerely,

Valdir Sabbaga Amato, Ph.D

Guest Editor

Claudia Brodskyn

Section Editor

Reviewer's Responses to Questions

**Key Review Criteria Required for Acceptance?**

**Methods**

-Are the objectives of the study clearly articulated with a clear testable hypothesis stated?

-Is the study design appropriate to address the stated objectives?

-Is the population clearly described and appropriate for the hypothesis being tested?

-Is the sample size sufficient to ensure adequate power to address the hypothesis being tested?

-Were correct statistical analysis used to support conclusions?

-Are there concerns about ethical or regulatory requirements being met?

Reviewer #1: This study aimed to investigate the incidence and persistence of Leishmania infection among HIV-infected individuals in an affected area, Trang Province, Southern Thailand. The study also identified associated risk factors.

We think the objectives proposed by the authors are perfectly articulated with the hypothesis or question that would motivate an expert on the subject.

Inclusion criteria were established. 

It is a cohort longitudinal study that enrolled 386 participants in the HIV clinic, Trang Hospital, who initially tested negative for Leishmania infection during 2015-2016, and 133 individuals initially tested positive for Leishmania infection. Thus, follow-up visits of 519 participants occurred during 2018-2019. 

Direct Agglutination Test (DAT) and PCR identified incident and persistent Leishmania infection cases.

It was calculated density and persistence of Leishmania infection. 

Cox proportional-hazards regression analyses were performed to assess risk factors for the incidence of Leishmania infection.

The proportional hazards assumption for Cox regression analysis was assessed using scaled Schoenfeld residuals. Hazard ratios (HRs) and 95% confidence intervals (CIs) were calculated for the associations, and P- values of 0.05 or less were considered statistically significant. All statistical analyses were conducted using STATA, version SE14. 

- We think the objectives are clearly articulated with a clear testable hypothesis.

- We think the study design is appropriate to address the stated objectives. 

- We think the population is clearly described and appropriated for the hypothesis tested.

- We think the sample size is adequate to ensure adequate power to address the hypothesis tested.

- We think the statistical analysis used supports the conclusions, too.

- There are not concerns about ethical or regulatory requirements.

About ethical or regulatory requirements: this study followed the ethical principles outlined in the Declaration of Helsinki. Ethical approval was obtained from the Institutional Review Board of the Royal Thai Army Medical Department (approval number: S034h/60). The written informed consent was obtained from all study participants before their inclusion in the study. Confidentiality and anonymity of the participants were ensured by assigning unique identification codes to the collected data.

Reviewer #2: Let me start by saying that I like the clear definitions provided in the methods on asymptomatic Leishmania infection, symptomatic VL, and seropositivity. The methods are described well enough, but are not entirely clear at some points. The objectives are mostly clear.

The thing I mostly have doubts on are the sample size, but this may be due to a misunderstanding in the methods (see minor comments below) on my part. While the total sample size seems large (n=643 baseline visit, n=519 follow-up), it seems that the authors stratify these groups to identify risk factors associated with incident Leishmania infection (n = 12). Am I correct in my understanding that the authors are comparing within the incident Leishmania group (n = 12) whether those with an age above 40 years old have a higher/lower hazard ratio than those below 40 years old? Thus, while the total n > 500, the n = 12 is for this analysis? It seems strange to me to compare within this group, rather than compare between the population-at-risk that did not have the event (so the n=374 ‘Negative’ group) and those that did have the event (the n=12 ‘Incident Leishmania infection’ cases).

In addition to this more major concern, there are a few things regarding the study design that are unclear and would benefit from some suggestions/comments to improve the clarity:

1. On Line 102, where part of the objectives are already mentioned, it would be best to clearly define what the risk factors are for. It seems to be for incident Leishmania infection, rather than VL development. This needs to be made clear at this sentence already.

2. In the small paragraph where the authors define seropositivity, seroconversion can be defined also.

3. Relating to comment #2, the authors could benefit from mentioning how the studied population relates to the definitions provided, already in the methods, instead of the results. For example, the authors mention in the results on L268 that individuals testing negative for Leishmania at the baseline visit are the population at risk for incident Leishmania infection. This can be related to the definitions in the methods section already.

4. I believe it would be helpful if the authors would adjust Figure 1 to highlight what exactly they are comparing and to what objectives it relates. 

5. In the results section in Table 4, “Crude Leishmania infection” is mentioned. This is not mentioned anywhere in the methods section. What is this definition?

6. In this study, the ‘asymptomatic Leishmania infection’ is defined as testing positive for a DAT or nPCR. While I do not necessarily disagree on whether these can be defined as asymptomatic or not, they do omit VL history in this definition. Do the authors have any idea on the clinical history of the participants? Are those with seropositivity past VL cases, or are they truly ‘resistant’ asymptomatic cases? If the authors have this data, it would be wise to report it.

7. The statistics for the baseline characteristics in Table 1 were not reported. What statistical test was used?

8. Arbitrary categories for variables were made to test for, such as age <= 40 or age >40. Why not analyse this as a continuous variable? If you did not find any statistical difference between age <=40 and >40, would you then adjust to age <=50 and >50 to find a statistical difference? This hypothetical scenario highlights why you should not analyse some continuous variables as a categorical variable unless there is biological reasoning for the threshold.

Reviewer #3: -Are the objectives of the study clearly articulated with a clear testable hypothesis stated?

Answer: Yes

-Is the study design appropriate to address the stated objectives?

Answer:Yes

-Is the population clearly described and appropriate for the hypothesis being tested?

Answer:Yes

-Is the sample size sufficient to ensure adequate power to address the hypothesis being tested?

Answer:Yes

-Were correct statistical analysis used to support conclusions?

Answer:Yes

-Are there concerns about ethical or regulatory requirements being met?

Answer: No

**Results**

-Does the analysis presented match the analysis plan?

-Are the results clearly and completely presented?

-Are the figures (Tables, Images) of sufficient quality for clarity?

Reviewer #1: Regarding the results, the study enrolled 386 participants in the HIV clinic, Trang Hospital, who initially tested negative for Leishmania infection during 2015-2016, and 133 individuals initially tested positive for Leishmania infection. Among the initially negative group, incident cases comprised one L. orientalis infection and 11 seropositive cases using DAT, resulting in a cumulative incidence of 3.1% and an incidence density of 10.38 per 1000 person- years. Five persistent cases comprised one L. donovani complex and four seropositive cases using DAT in the initially positive group, with a cumulative persistence rate of 3.7% and a persistence density of 12.85 per 1000 person-years 

-Does the analysis presented match the analysis plan? We think yes. 

- Are the results clearly and completely presented? We think yes, too

- Are the figures (Tables, Images) of sufficient quality for clarity? Yes

Reviewer #2: The more major concern I have with the analysis is similar as mentioned above under the ‘Methods’ comments. It is unclear whether the bivariate/multivariate Cox proportional-hazards regression was performed on only the n=12 incident Leishmania cases, or on the full population-at-risk?

In addition to this, I have a few minor suggestions to improve the clarity of the results section:

1. Table 1 currently presents results stratified by follow-up status. This makes the table hard to interpret, and is really only interesting to validate whether the results are representable. As a suggestion, I would move this table to a supplementary file, while presenting the participant characteristics for either all patients or for only those patients that were followed-up in the main results as Table 1.

2. In Table 1, it seems that statistical tests were only performed comparing within-group based on follow-up status? So the authors only compared between a ‘Yes’ and ‘No’ follow-up status for negative Leishmania infection, and separately for positive Leishmania infection? If this is correct, it would be beneficial to adapt this to comparing negative Leishmania infection to positive Leishmania infection, either at baseline or only for those followed-up.

3. In Table 1, please define the units of the variables. For example, is age represented as n (%)?

4. In Table 4, please define Crude Leishmania infection.

5. Am I correct in my understanding that there were no statistical tests for Occupation, Intravenous drug user, and Recreational drug user, in Table 4?

Reviewer #3: -Does the analysis presented match the analysis plan?

Answer: yes

-Are the results clearly and completely presented?

Answer: Yes

-Are the figures (Tables, Images) of sufficient quality for clarity?

Answer:no

**Conclusions**

-Are the conclusions supported by the data presented?

-Are the limitations of analysis clearly described?

-Do the authors discuss how these data can be helpful to advance our understanding of the topic under study?

-Is public health relevance addressed?

Reviewer #1: This study sheds light on the incidence and persistence of Leishmania infection among HIV-infected individuals in Trang Province, Southern Thailand. It has immediate implications for healthcare providers, policymakers, and public health officials, aiding them in implementing targeted interventions and preventive measures to reduce the impact of leishmaniasis among HIV-infected individuals. 

- Are the conclusions supported by the data presented? Yes. 

- Are the limitations of analysis clearly described? Yes

- Do the authors discuss how these data can be helpful to advance our understanding of the topic under study? Yes. 

- Is public health relevance addressed. Yes.

Reviewer #2: With regard to public health relevance, this is clearly addressed. The authors can, however, be a bit more specific on how this data can be used to advance public health measures, with specific measures linked to the outcomes.

However, the conclusions are not always supported by the data presented. This relates to my major concern above.

Reviewer #3: -Are the conclusions supported by the data presented?

Answer: no

-Are the limitations of analysis clearly described?

Answer: no

-Do the authors discuss how these data can be helpful to advance our understanding of the topic under study?

Answer: yes

-Is public health relevance addressed

Answer:yes

**Editorial and Data Presentation Modifications?**

Reviewer #1: WE COMMENT AND SUGGEST:

- In all the text there is a lack of standardization of the nomenclature of Leishmania. 

- Pag 18, line 315. Among the positive cases, 11 individuals tested positive for Leishmania antibodies using DAT alone. 

DAT appears to the best serological test for leishmaniasis both in terms of sensitivity and specificityfalse. But this reaction can cross with other diseases and this must be taken into consideration, especially considering immunosuppressed patients, therefore more vulnerable to infections. Furthermore, it is known that there are cases of Trypanosoma infection in Thailand. 

On the other hand, it is known that there is a window of false positivity or negativity in both tests used, DAT and PCR. 

 Therefore, I think it would be appropriate to comment on these situations. 

To give more credibility to the results, it would be good to clarify whether there was more than one technician who read the DAT.

Pag. 20, line 342. Table 3. Patient number 5. There should be an effort by the authors to identify the Leishmania species in both the baseline diagnosis and follow-up to prove that the persistence of the infection was caused by the same species, reducing the possibility of a new infection.

Reviewer #2: Minor Revision

Reviewer #3: (No Response)

**Summary and General Comments**

Reviewer #1: - The claims are properly placed in the context of the previous literature. 

- The authors have treated the literature fairly.

- The data and analyses fully support the claims.

- The results open perspectives for new researches as a multicentric in Thailand related to HIV infection patients.

- This is a paper outstanding in coinfection HIV/Leishmaniasis that inspires new questions about.

- The details of the methodology are sufficient to allow the experiments to be reproduced. They followed the checklist of items of STROBE guidelines for reporting observational studies.

- The manuscript is well organized and written clearly enough to be accessible to non-specialists.

Reviewer #2: The study by Bualert et al. focuses on the incidence of asymptomatic Leishmania infection in people living with HIV in a province of southern Thailand, and assesses several risk factors for Leishmania infection. This study in the most neglected subset (HIV-positive individuals) of already neglected patients is of crucial importance. One of the key strengths is the longitudinal design, with a baseline visit and a follow-up visit around 2-3 years later. I also like that they provide clear explanations on their definitions of asymptomatic Leishmania infection and symptomatic VL. However, the study results are not presented clearly at times, and can perhaps be clarified (see feedback/comments above and below).

Major comments:

1. The thing I mostly have doubts on are the sample size, but this may be due to a misunderstanding in the methods (see minor comments below) on my part. While the total sample size seems large (n=643 baseline visit, n=519 follow-up), it seems that the authors stratify these groups to identify risk factors associated with incident Leishmania infection (n = 12). Am I correct in my understanding that the authors are comparing within the incident Leishmania group (n = 12) whether those with an age above 40 years old have a higher/lower hazard ratio than those below 40 years old? Thus, while the total n > 500, the n = 12 is for this analysis? It seems strange to me to compare within this group, rather than compare between the population-at-risk that did not have the event (so the n=374 ‘Negative’ group) and those that did have the event (the n=12 ‘Incident Leishmania infection’ cases). If this is truly the case, I would suggest to re-do this analysis.

Minor comments:

1. In the Author Summary, on L68-L69, the authors state that “Even individuals who initially tested negative for Leishmania are not immune to infection’. Is this not already obvious? This study does not reveal this, and no researcher thinks that is the case. Consider to rewrite or remove this sentence.

2. In line with comment #1, the authors write on L363-L367 that individuals with an initial negative status are consider to be at a lower risk of Leishmania infection? This is definitely not considered. A negative infection test does not mean an individual is at a lower risk of future infection, it only states a person is not infected at that time.

3. It is not clear what is meant in L40-L42 with ‘reveals little information’.

4. On L44-L45, please define what the risk factors are associated with. In this case, it seems to be incident Leishmania infection, so the risk of contracting a Leishmania infection. This is important to define, as risk factors for infection can be different from disease development.

5. On L80-81, the authors mention that leishmaniasis is prevalent globally in the Indian subcontinent and East Africa. What about South-America, or Asia, such as Thailand where the study situates?

6. The introduction does not touch upon any already known risk factors for incident Leishmania infection, even on other continents and in other regions. Consider to introduce this already.

7. On Line 102, where part of the objectives are already mentioned, it would be best to clearly define what the risk factors are for. It seems to be for incident Leishmania infection, rather than VL development. This needs to be made clear at this sentence already.

8. In the small paragraph where the authors define seropositivity, seroconversion can be defined also.

9. Relating to comment #2, the authors could benefit from mentioning how the studied population relates to the definitions provided, already in the methods, instead of the results. For example, the authors mention in the results on L268 that individuals testing negative for Leishmania at the baseline visit are the population at risk for incident Leishmania infection. This can be related to the definitions in the methods section already.

10. I believe it would be helpful if the authors would adjust Figure 1 to highlight what exactly they are comparing and to what objectives it relates. 

11. In the results section in Table 4, “Crude Leishmania infection” is mentioned. This is not mentioned anywhere in the methods section. What is this definition?

12. In this study, the ‘asymptomatic Leishmania infection’ is defined as testing positive for a DAT or nPCR. While I do not necessarily disagree on whether these can be defined as asymptomatic or not, they do omit VL history in this definition. Do the authors have any idea on the clinical history of the participants? Are those with seropositivity past VL cases, or are they truly ‘resistant’ asymptomatic cases? If the authors have this data, it would be wise to report it.

13. On L162-L163, the authors mention that the individuals continued to harbor an active and detectable Leishmania infection through-out the follow-up period. Did they consistently do periodic testing from the baseline visit until the follow-up visit, or do they mean that participants that tested positive on the baseline visit still tested positive years later on the follow-up visit? This is an important difference.

14. In the methods section, on L163-L165, the authors write about their defined persistent cases not achieving complete cure. However, there is no mention of treatment, nor of past VL history. Were these participants ever treated? Or could it be a re-infection rather than persistence? This is not adequately discussed and there is too little information provided to speculate.

15. The statistics for the baseline characteristics in Table 1 were not reported. What statistical test was used?

16. In Table 1, it seems that statistical tests were only performed comparing within-group based on follow-up status? So the authors only compared between a ‘Yes’ and ‘No’ follow-up status for negative Leishmania infection, and separately for positive Leishmania infection? If this is correct, it would be beneficial to adapt this to comparing negative Leishmania infection to positive Leishmania infection, either at baseline or only for those followed-up.

17. Table 1 currently presents results stratified by follow-up status. This makes the table hard to interpret, and is really only interesting to validate whether the results are representable. As a suggestion, I would move this table to a supplementary file, while presenting the participant characteristics for either all patients or for only those patients that were followed-up in the main results as Table 1.

18. In Table 1, please define the units of the variables. For example, is age represented as n (%)?

19. The paragraph ‘Investigation of persistent Leishmania infection’ on L285 seems to be about the positive Leishmania infection group, and not the persistent group. Reconsider the title.

20. In Table 4, please define Crude Leishmania infection.

21. Am I correct in my understanding that there were no statistical tests for Occupation, Intravenous drug user, and Recreational drug user, in Table 4?

22. In the discussion, the authors compare incidence rates of this study to those in other settings such as North-West Ethiopia. However, these are different incidence rates. This present study defines the incidence of Leishmania infection, while those study assess leishmaniasis cases (so disease incidence). Consider to mention this discrepancy.

23. Arbitrary categories for variables were made to test for, such as age <= 40 or age >40. Why not analyse this as a continuous variable? If you did not find any statistical difference between age <=40 and >40, would you then adjust to age <=50 and >50 to find a statistical difference? This hypothetical scenario highlights why you should not analyse some continuous variables as a categorical variable unless there is biological reasoning for the threshold.

24. Relating to comment #23, since you identify age as a significant associated factor with a categorical arbitrary split, consider to include an analysis where you use it as a continuous variable.

25. There is no reference provided for L445-L446 where the authors state that individuals with lower CD4 counts are at a higher risk of Leishmania infection. Most of the literature on CD4 counts states it is a risk factor particularly for VL development (and subsequent relapse), and not particularly for infection. This is an important distinction that the authors sometimes do not make.

Reviewer #3: In this manuscript, the authors evaluated the incidence and persistence of asymptomatic Leishmania infection in people living with HIV in the southern region of Thailand. The topic is very interesting and relevant for understanding Leishmania and HIV co-infection. The manuscript has very relevant information, however some points need to be made clearer to make the work more robust. Below, I send my review and sugestions.

Introduction:

1) I suggest include a paragraph regarding the sensivitiy and sensbility of the methosd used to detect Leishmania infection.

2) I suggest include new lasted others information about coinfection Leishmania and HIV, mainly those used different methodology to detect Leishmani infection. 

Methodologgy

1) I suggest includ anny other methodoly to detect antibodies, not only DAT. 

2) Regarding DAT, what the Leishmania specie and strain used as antigens? It is an importan point, due to reaction with the Leishmania species. 

Result

1) In the table 1, I suggest counting CD4 lymphocytes at three levels (<200, >200 and <500, >500) to assess whether there was a difference in incidence and persistence of infection.

2) Regarding the identification of infection by L. (V.) lainsoni, it would be essential to clarify the epdemiology of this patient, as this species, as well as the others in the subgenus Viannia, only occur in Latin America. It would be essential to review this result, as there may have been an error in identifying the species or the patient may have become infected when traveling to an area where L. lainsoni occurs.

Discussion:

An important point to take into consideration in the discussion is the methodology used to determine infection. It is known that the methodologies have different sensitivity. Another point to be taken into consideration in the discussion refers to the specificity of the DAT. There is a description in the literature about cross-reactivity of serological methods for diagnosing leishmaniasis with other diseases, such as tuberculosis. Cunha et al obtained different frequency percentages, using different methodologies to detect Leishmania infection in people living with HIV.(Cunha MA, Celeste BJ, Kesper N et al. Frequency of Leishmania spp. infection among HIV-infected patients living in an urban area in Brazil: a cross-sectional study. BMC Infect Dis. 2020 Nov 25;20(1):885). I suggest consider this information in the discussion.

PLOS authors have the option to publish the peer review history of their article (what does this mean?). If published, this will include your full peer review and any attached files.

Reviewer #1: Yes: Raimunda Nonata Ribeiro Sampaio

Reviewer #2: Yes: Nicky de Vrij

Reviewer #3: No
---

## [Decision Letter · Decision Letter 1]

14 Aug 2024

Dear Assoc. Prof. Piyaraj,

Thank you very much for submitting your manuscript "Incidence and persistence of asymptomatic Leishmania infection among HIV-infected patients in Trang province, southern Thailand: a cohort study" for consideration at PLOS Neglected Tropical Diseases. As with all papers reviewed by the journal, your manuscript was reviewed by members of the editorial board and by several independent reviewers. In light of the reviews (below this email), we would like to invite the resubmission of a significantly-revised version that takes into account the reviewers' comments. 

We cannot make any decision about publication until we have seen the revised manuscript and your response to the reviewers' comments. Your revised manuscript is also likely to be sent to reviewers for further evaluation.

Sincerely,

Valdir Sabbaga Amato, Ph.D

Academic Editor

Claudia Brodskyn

Section Editor

Reviewer's Responses to Questions

**Key Review Criteria Required for Acceptance?**

**Methods**

-Are the objectives of the study clearly articulated with a clear testable hypothesis stated?

-Is the study design appropriate to address the stated objectives?

-Is the population clearly described and appropriate for the hypothesis being tested?

-Is the sample size sufficient to ensure adequate power to address the hypothesis being tested?

-Were correct statistical analysis used to support conclusions?

-Are there concerns about ethical or regulatory requirements being met?

Reviewer #1: This study aimed to investigate the incidence and persistence of Leishmania infection among HIV-infected individuals in an affected area, Trang Province, Southern Thailand. The study also identified associated risk factors. We think the objectives proposed by the authors are perfectly articulated with the hypothesis or question that would motivate an expert on the subject.

Inclusion criteria were established.

It is a cohort longitudinal study that enrolled 373 participants in the HIV clinic, Trang Hospital, who initially tested negative for Leishmania infection during 2015-2016, and 133 individuals initially tested positive for Leishmania infection. Thus, follow-up visits of 506 participants occurred during 2018-2019.

Direct Agglutination Test (DAT) and PCR identified incident and persistent Leishmania infection cases.

It was calculated density and persistence of Leishmania infection.

Cox proportional-hazards regression analyses were performed to assess risk factors for the incidence of Leishmania infection.

The proportional hazards assumption for Cox regression analysis was assessed using scaled Schoenfeld residuals. Hazard ratios (HRs) and 95% confidence intervals (CIs) were calculated for the associations, and P- values of 0.05 or less were considered statistically significant. All statistical analyses were conducted using STATA, version SE18.

About ethical or regulatory requirements: this study followed the ethical principles outlined in the Declaration of Helsinki. Ethical approval was obtained from the Institutional Review Board of the Royal Thai Army Medical Department (approval number: S034h/60). The written informed consent was obtained from all study participants before their inclusion in the study. Confidentiality and anonymity of the participants were ensured by assigning unique identification codes to the collected data.

Reviewer #2: The authors addressed my concerns sufficiently.

Reviewer #3: - Are the objectives of the study clearly articulated with a clear testable hypothesis stated? Yes

-Is the study design appropriate to address the stated objectives? Yes

-Is the population clearly described and appropriate for the hypothesis being tested? Yes

-Is the sample size sufficient to ensure adequate power to address the hypothesis being tested? Yes

-Were correct statistical analysis used to support conclusions? Yes

-Are there concerns about ethical or regulatory requirements being met? Yes

**Results**

-Does the analysis presented match the analysis plan?

-Are the results clearly and completely presented?

-Are the figures (Tables, Images) of sufficient quality for clarity?

Reviewer #1: Regarding the results, the study enrolled 373 participants in the HIV clinic, Trang Hospital, who initially tested negative for Leishmania infection during 2015-2016, and 133 individuals initially tested positive for Leishmania infection. Among the initially negative group, incident cases comprised one L. orientalis infection and 11 seropositive cases using DAT, resulting in a cumulative incidence of 3.2% and an incidence density of 10.38 per 1000 person- years. Increasing age was a significant predictor of the incidence of Leishmania infection. Five persistent cases comprised one L. donovani complex and four seropositive cases using DAT in the initially positive group, with a cumulative persistence rate of 3.7% and a persistence density of 12.85 per 1000 person-years.

Reviewer #2: The authors addressed my concerns sufficiently.

Reviewer #3: -Does the analysis presented match the analysis plan? Yes

-Are the results clearly and completely presented? No

-Are the figures (Tables, Images) of sufficient quality for clarity? No

**Conclusions**

-Are the conclusions supported by the data presented?

-Are the limitations of analysis clearly described?

-Do the authors discuss how these data can be helpful to advance our understanding of the topic under study?

-Is public health relevance addressed?

Reviewer #1: The conclusions are supported by the data and the limitations clearly described. The authors discuss how these data can be helpful to advance our understanding of the topic under study.This study sheds light on the incidence and persistence of Leishmania infection among HIV-infected individuals in Trang Province, Southern Thailand. It has immediate implications for healthcare providers, policymakers, and public health officials, aiding them in implementing targeted interventions and preventive measures to reduce the impact of leishmaniasis among HIV-infected individuals.

Reviewer #2: The authors addressed my concerns sufficiently.

Reviewer #3: -Are the conclusions supported by the data presented? Yes

-Are the limitations of analysis clearly described? Yes

-Do the authors discuss how these data can be helpful to advance our understanding of the topic under study? Yes

-Is public health relevance addressed? Yes

**Editorial and Data Presentation Modifications?**

Reviewer #1: I have no suggestions for modifications and I think this manuscript should be accepted.

Reviewer #2: Accept.

Reviewer #3: (No Response)

**Summary and General Comments**

Reviewer #1: No comments

Reviewer #2: The authors addressed my concerns sufficiently.

Reviewer #3: The way the results are presented is confusing. For example, regarding Leishmania infection in the baseline, it is not clear whether the 164 positive cases were due to antibody detection or Leishmania DNA detection. I suggest making this information clearer. I think it should show how many are serologically positive and how many of these had a positive PCR test, initially.

Regarding the persistence of the infection, what is the methodology, DAT only? How many DAT positives were tested by PCR and how many were positive by PCR?

Regarding Table 2. These results are completely confusing. I don't understand. I suggest you review this information. Apparently, of the 133 patients with positive DAT in the Baseline, only 7 had Leishmania DNA detected. In 98, DAT was performed and 98 had positive DAT. In another 35, only sequencing was performed and no PCR was performed?? According to what is shown in the table, 7 patients showed positive DAT and positive nPCR, is that it?

The authors continue to insist on the infection by L. lainsoni, however the results presented do not confirm the infection by L. lainsoni. I believe that this information is biased, since only L. lainsoni was detected using the ITS-1 target and even when using kDNA (9, which has more copies) the PCR was negative. This information is not real.

PLOS authors have the option to publish the peer review history of their article (what does this mean?). If published, this will include your full peer review and any attached files.

Reviewer #1: Yes: Raimunda Nonata Ribeiro Sampaio

Reviewer #2: Yes: Nicky de Vrij

Reviewer #3: No
---

## [Editor Report · Decision Letter 2]

27 Sep 2024

Dear Assoc. Prof. Piyaraj,

We are pleased to inform you that your manuscript 'Incidence and persistence of asymptomatic Leishmania infection among HIV-infected patients in Trang province, southern Thailand: a cohort study' has been provisionally accepted for publication in PLOS Neglected Tropical Diseases.

Best regards,

Abhay R Satoskar

Section Editor

Claudia Brodskyn

Section Editor

---

## [Editor Report · Acceptance letter]

2 Oct 2024

Dear Assoc. Prof. Piyaraj,

We are delighted to inform you that your manuscript, "Incidence and persistence of asymptomatic Leishmania infection among HIV-infected patients in Trang province, southern Thailand: a cohort study," has been formally accepted for publication in PLOS Neglected Tropical Diseases.

Best regards,

Shaden Kamhawi

co-Editor-in-Chief

Paul Brindley

co-Editor-in-Chief
